# Ascorbate-Glutathione Cycle Genes Families in Euphorbiaceae: Characterization and Evolutionary Analysis

**DOI:** 10.3390/biology12010019

**Published:** 2022-12-22

**Authors:** Douglas Jardim-Messeder, Ygor de Souza-Vieira, Lucas Corrêa Lavaquial, Daniela Cassol, Vanessa Galhego, Gabriel Afonso Bastos, Thais Felix-Cordeiro, Régis Lopes Corrêa, Marcel Zámocký, Márcia Margis-Pinheiro, Gilberto Sachetto-Martins

**Affiliations:** 1Laboratório de Genômica Funcional e Transdução de Sinal, Departamento de Genética, Universidade Federal do Rio de Janeiro, Avenida Carlos Chagas Filho 373, Cidade Universitária, Rio de Janeiro 21941-590, Brazil; 2Laboratório de Biologia Molecular de Plantas, Instituto de Bioquímica Médica Leopoldo de Meis, Universidade Federal do Rio de Janeiro, Avenida Carlos Chagas Filho 373, Cidade Universitária, Rio de Janeiro 21941-590, Brazil; 3Instituto de Biología Integrativa de Sistemas (I2SysBio), Consejo Superior de Investigaciones Científicas (CSIC)—Universitat de València, Paterna, 46980 Valencia, Spain; 4Laboratory of Phylogenomic Ecology, Institute of Molecular Biology, Slovak Academy of Sciences, Dúbravská cesta 21, 84551 Bratislava, Slovakia; 5Department of Chemistry, Institute of Biochemistry, University of Natural Resources and Life Sciences, Muthgasse 18, 1190 Vienna, Austria; 6Laboratório de Genômica Funcional de Plantas, Departamento de Genética, Universidade Federal do Rio Grande do Sul, Avenida Bento Gonçalves 9500, Agronomia, Porto Alegre 90650-001, Brazil; 7Centro de Biotecnologia, Universidade Federal do Rio Grande do Sul, Avenida Bento Gonçalves 9500, Agronomia, Porto Alegre 90650-001, Brazil

**Keywords:** reactive oxygen species, ascorbate peroxidase, monodehydroascorbate reductase, dehydroascorbate reductase, glutathione reductase, ascorbate-glutathione cycle

## Abstract

**Simple Summary:**

The climatic changes pose important threats to agriculture. Plants are sessile organisms and need to cope with different environmental conditions throughout their lifespan. Most responses involve different signaling molecules, including reactive oxygen species (ROS). Due to their high reactivity, ROS are also an oxidative threat to the cell. Consequently, plant cells display elaborated defenses against oxidative stress. The ascorbate-glutathione cycle is the main antioxidant pathway in photosynthetic organisms and is composed by the enzymes ascorbate peroxidase (APX), monodehydroascorbate reductase (MDAR), dehydroascorbate reductase (DHAR), and glutathione reductase (GR). Here, the APX, MDAR, DHAR and GR genes from castor bean, cassava, jatropha and rubber tree were identified and classified. This classification allowed the prediction of their subcellular localization within plant cells, such as cytosol, peroxisomes, chloroplasts, and mitochondria. Our analysis also contributes to understanding the evolutionary history of these genes. The expression pattern of ascorbate-glutathione cycle genes in castor bean submitted to drought reveals changes in leaves and roots. Altogether, these data contribute to uncovering the regulation of ROS metabolism during stress response in castor bean, which is highly tolerant to drought.

**Abstract:**

Ascorbate peroxidase (APX), Monodehydroascorbate Reductase (MDAR), Dehydroascorbate Reductase (DHAR) and Glutathione Reductase (GR) enzymes participate in the ascorbate-glutathione cycle, which exerts a central role in the antioxidant metabolism in plants. Despite the importance of this antioxidant system in different signal transduction networks related to development and response to environmental stresses, the pathway has not yet been comprehensively characterized in many crop plants. Among different eudicotyledons, the Euphorbiaceae family is particularly diverse with some species highly tolerant to drought. Here the APX, MDAR, DHAR, and GR genes in *Ricinus communis*, *Jatropha curcas*, *Manihot esculenta*, and *Hevea brasiliensis* were identified and characterized. The comprehensive phylogenetic and genomic analyses allowed the classification of the genes into different classes, equivalent to cytosolic, peroxisomal, chloroplastic, and mitochondrial enzymes, and revealed the duplication events that contribute to the expansion of these families within plant genomes. Due to the high drought stress tolerance of *Ricinus communis,* the expression patterns of ascorbate-glutathione cycle genes in response to drought were also analyzed in leaves and roots, indicating a differential expression during the stress. Altogether, these data contributed to the characterization of the expression pattern and evolutionary analysis of these genes, filling the gap in the proposed functions of core components of the antioxidant mechanism during stress response in an economically relevant group of plants.

## 1. Introduction

During their evolution, aerobic organisms acquired the capacity to use the abundant atmospheric O_2_ to oxidize organic compounds to obtain chemical energy in a highly efficient manner. Paradoxically, the monovalent reduction of molecular oxygen produces different partially reduced compounds, known as reactive oxygen species (ROS), which can react with numerous biological molecules and cause serious cellular damage [1]. Consequently, aerobic organisms developed enzymatic and nonenzymatic mechanisms to control ROS levels and prevent oxidative damage [2]. In addition, mechanisms evolved in these organisms allow the use of ROS in different signaling pathways, related to development and stress response [3]. Plant growth and, consequently, worldwide agricultural production can be affected by different environmental conditions, which have, as a primary response, the increase of ROS production, which acts in stress response pathways [4].

Peroxidases, in addition to superoxide dismutases and catalases, are an important group of antioxidant enzymes [5]. Among them, ascorbate peroxidase (APX, EC 1.11.1.11) is recognized as the main antioxidant enzyme in non-animal organisms, such as higher plants and chlorophytes [6], red algae [7], and protists [8]. APX displays extraordinarily high specificity in using ascorbate as the electron donor to reduce hydrogen peroxide. This is accomplished through a set of reactions known as the “ascorbate–glutathione cycle” (AsA-GSH) [9]. In this pathway (Figure 1a), ascorbate acts as a reducing substrate to eliminate hydrogen peroxide, and, posteriorly, is recycled through NAD(P)H or glutathione oxidation, by monodehydroascorbate reductase (MDAR, EC 1.6.5.4) and dehydroascorbate reductase (DHAR, EC 1.8.5.1), respectively. Finally, glutathione reductase (GR, EC 1.8.1.7) reduces the oxidized glutathione through NAD(P)H oxidation.

In higher plants, small multigene families encode the APX family, and their different isoforms are classified according to their subcellular localization, which is determined primarily by organelle-specific target sequences, as well as transmembrane domains in N- or C- terminal regions [10]. In angiosperms, the APX family is composed of genes that encode cytosolic (cAPX), peroxisomal (pAPX), chloroplastic (chlAPX), and mitochondrial (mitAPX) isoforms [11]. The MDAR family is divided into three phylogenetic groups named I, II, and III, which correspond to chloroplastic/mitochondrial (chl/mitMDAR), peroxisomal (pMDAR), and cytosolic/peroxisomal (c/pMDAR) isoforms, respectively. In Brassicaceae, a specific subclass III corresponding to MDAR enzymes targeted only to the cytosol (cMDAR) is observed [12]. The DHAR family is composed of two groups targeted to cytosol (cDHAR) and chloroplast (chlDHAR) [13,14]. Two classes of GR proteins have been also reported, based on the presence of an N-terminal extension. Class I corresponds to shorter cytosolic enzymes (cGR), while class II corresponds to larger enzymes targeted to both chloroplast and mitochondria (chl/mitGR) [15].

The analysis of eukaryotic and prokaryotic genomes suggests that GR and the glutathione metabolism appeared early during evolution, possibly due to the change from a reducing to an oxidizing atmosphere in the primitive Earth [16]. Thus, the GR family is found in most organisms, except for some strict anaerobes and primitive Archaebacteria species that do not use glutathione as an antioxidant molecule [16]. On the other hand, the APX, MDAR, and DHAR families are found in photosynthetic eukaryotes, but not in Glaucophyte or Cyanobacteria, suggesting that the emergence of the ASA-GSH cycle occurred more recently, selectively in photosynthetic eukaryotes, after the plastid acquisition and Glaucophyte divergence, as a response to the use of ascorbate as antioxidant molecule [17]. In addition, APX, MDAR, and DHAR appeared initially as chloroplastic enzymes, and the posterior emergence of multiple isoforms targeted to different subcellular locations occurred during the evolution of the first land plants, allowing important evolutionary advantages to cope with harsh conditions of terrestrial life [18].

The expression of genes related to the AsA-GSH cycle is highly regulated during different abiotic or biotic stress, alleviating oxidative damage and being essential components of cell signaling pathways triggering adaptive responses [19,20]. Although the AsA-GSH cycle genes have been studied, functional data are still scarce for many species of economic interest. 

The Euphorbiaceae is a taxonomic plant family with high morphological diversity. In addition to species of great importance for human consumption, such as cassava (*Manihot esculenta*), products derived from some species of the Euphorbiaceae family are also widely used in the textile and chemical industry, such as rubber tree latex (*Hevea brasiliensis*), and the oil derived from the castor bean (*Ricinus communis*) or jatropha (*Jatropha curcas*) seeds. 

Among these species, castor bean is especially interesting. It is easy to grow in unfavorable conditions, such as warm climates, and is highly tolerant to limited water supply [21,22,23]. The castor bean presents high oil content in the seeds [22], being composed of 90% of ricinoleic acid, which is useful in different industrial sectors, being an ecologically friendly additive [24,25,26]. As expected, castor bean seeds yield and oil quality are adversely affected by abiotic stresses, in which the response is strongly regulated by ROS signaling pathways. 

In castor bean, APX protein has been first purified from peroxisomal membranes [27] and total APX activity was demonstrated to be increased in plants submitted to gamma radiation exposure [28] and drought stress [29]. Similarly, in jatropha, the APX activity is also increased during drought stress [29], and the overexpression of cDHAR gene from jatropha in *Nicotiana benthamiana* enhances the tolerance to hydrogen peroxide, salt, and polyethylene glycol (PEG) stress [30]. 

Despite the importance of ROS metabolism in plant stress response mechanisms, not all AsA-GSH cycle genes have been identified in Euphorbiaceae species. Here, we identified the APX, MDAR, DHAR, and GR genes in castor bean, cassava, jatropha, and rubber tree genomes. The phylogenetic analysis allowed us to classify the genes into different subfamilies. A comprehensive analysis of chromosomal distribution, duplication events that contribute to the expansion of these families, protein structure prediction, and conservation of amino acids was also performed. To explore the castor bean response to drought, the expression profile of APX, MDAR, DHAR, and GR genes was evaluated in leaves and roots. Differential expression of AsA-GSH cycle genes was observed mainly in leaves, where they are generally repressed during initial stress. In addition, we analyzed the promoter cis-acting elements and putative microRNA targets that could regulate the expression of these genes. Altogether, these data contribute to their functional characterization and could increase the understanding of the core components of the antioxidant mechanism involved in the efficient adaptation of castor bean to drought stress.

Drought stress is recognized as one of the most important abiotic factors limiting global food production. The exposure to drought leads to relevant yield losses annually, a situation intensified by climate changes and the freshwater-supply shortage [31,32]. To meet the demands of food security of an increasing world population, one of the greatest challenges of the century is a “second green revolution” to enhance crop yield even in adverse growing conditions [33,34]. 

## 2. Materials and Methods

### 2.1. Retrieval of APX, MDAR, DHAR, and GR Proteins

APX, MDAR, DHAR, and GR protein sequences of castor bean, cassava, jatropha, and rubber tree were retrieved from Phytozome v13 and Dicots Plaza 4.0 databases through BLASTp tool using sequences from rice (*Oryza sativa*) and arabidopsis (*Arabidopsis thaliana*) as bait (Appendix A), and a minimum threshold cut-off of e−20. Sequences were checked by reverse BLASTp in NCBI, and Pfam analysis to confirm the presence of conserved domains [35].

### 2.2. Phylogenetic Analysis

Amino acid sequences of APX, MDAR, DHAR, and GR proteins from arabidopsis, rice, castor bean, cassava, jatropha, and rubber tree were aligned using Multiple Sequence Comparison by Log Expectation tool (MUSCLE) [36]. The phylogenetic tree was made individually for each family using the neighbor-joining method under the best model selection in MEGA 11 [37] with 1000 replicates of bootstrap statistics. 

### 2.3. Chromosomal Positions and Synthenic Analysis

The gene location on the *Ricinus communis* and *Manihot esculenta* chromosomes was shown by Circos [38]. Detection of putative gene duplication events was done with MCScanX (E-value 1 × 10^−10^) in each genome and with a comparison between both genomes and visualized using Advanced Circos of TBtools software v1.098769 [39]. Tandem duplication events were defined as two or more homologous genes located on a chromosomal region within 200 kb [40]. Collinearity between *R. communis* and *M. esculenta* genomes was done with MCScanX (E-value 1 × 10^−10^) and visualized by TBtools v1.098769 [39].

### 2.4. Calculation of Ka/Ks and Divergence Time

The nucleotide and amino acid sequences of duplicated gene pairs were aligned and were estimated the number of non-synonymous substitutions per non-synonymous site (Ka), synonymous substitutions per synonymous site (Ks), and Ka/Ks ratio using KaKs_Calculator 2.0 software [41]. The divergence time was calculated according to T = Ks/(2 × 8.1 × 10^−9^) MYA for vascular plants [42].

### 2.5. Structural Analysis of APX, MDAR, DHAR, and GR Proteins

Molecular weight (MW), isoelectric point (pI), and GRAVY (grand average of hydropathy) from APX, MDAR, DHAR, and GR proteins were investigated through the ProtParam tool [43]. The conserved motifs in amino acid sequences were analyzed using MEME (Multiple Em for Motif Elicitation) (http://meme-suite.org/) (accessed on 14 January 2022) using the following parameters: number of motifs 1–15 and motif width of 5–50 [44]. Prediction of three-dimensional models was performed by Alphafold Protein Structure Database and the structural alignment was performed by Chimera v1.14 software [45]. To compare the primary sequence among arabidopsis, castor bean, cassava, jatropha, and rubber tree proteins, the translated sequences were aligned with MUSCLE tool [36] and submitted to the boxshade interface. 

### 2.6. Prediction of Potential Cis-Regulatory Elements

The putative regulatory region (1000 bp upstream from the translation start codon) of APX, MDAR, DHAR, and GR genes from arabidopsis, castor bean, cassava, jatropha, and rubber tree were retrieved, and the presence of cis-regulatory elements was identified by Plant Promoter Analysis Navigator from PlantPAN 3.0 database [46]. 

### 2.7. RNAseq and Expression Pattern Analysis

The expression data of castor bean APX, MDAR, DHAR, and GR genes in different tissues was obtained previously [47], while the expression profile of castor bean leaf and roots in response to drought (−0.5, −1.0, and −1.5 MPa) were retrieved from RNASeq experiment (NCBI GEO submission SUB3009450, bioproject PRJNA401329). The expression data were expressed in heat maps using a Log2 scale with relative values. The putative network interaction partners of APX proteins were predicted by STRING database [48] with default settings taking into account known interactions and predicted interactions. The expression profile of the interaction network was analyzed by VIA COMPLEX software V.1.0. [49].

### 2.8. Plants, Growth Conditions, and Drought Stress Experimental Design

The castor bean plants were sown in 15 L plastic pots with sandy loam soil. Plants were grown under continuous irrigation and natural photoperiod until the complete expansion of the third pair of leaves (approximately 2 months) when the drought treatment started. A suspension of irrigation protocol was employed. The plants were divided randomly into two groups: a control group in which irrigation was continued, and a group in which irrigation was suspended until a water potential of −0.5, −1.0, or −1.5 MPa was reached. Water potential was measured daily at pre-dawn with a Scholander-type pressure chamber. Six plants were used for each condition in the experiment. The water potential parameters were statistically analyzed by One-way ANOVA (Nonparametric) through Tukey post-test using GraphPad Prism 5. The plants samples were collected at the same time. The tissues were immediately frozen in liquid nitrogen and stored at −80 °C until processing.

### 2.9. Total RNA Isolation and Quantitative PCR (RT-qPCR) Analysis

Frozen samples were grounded in liquid nitrogen. Total RNA was isolated from 100 mg of each sample using the RNeasy Plant Mini Kit (Qiagen^®^, Mettmann, Germany), following the manufacturer’s instructions. The RNA purity and concentration were determined using a Nanodrop™ ND-1000 spectrophotometer (Thermo Scientific, Waltham, MA, USA) and RNA integrity was verified in a 1% agarose gel electrophoresis. The samples were treated with DNAase (Invitrogen^®^, Waltham, MA, USA) according to the manufacturer’s instructions to remove the eventual genomic DNA contamination and complementary DNA (cDNA) was synthesized from 1 μg of total RNA using the SuperScript III Reverse Transcriptase (Invitrogen^®^) and a 24-polyTV primer (Invitrogen^®^). After synthesis, cDNAs were diluted 50 times in sterile water for use in PCR reaction. 

All reactions were repeated four times, and expression data analyses were performed after comparative quantification of the amplified products using the 2^−ΔΔCt^ method [50,51]. RT-qPCR reactions were performed in 7500 Fast Real-Time PCR System (Applied Biosystems, Waltham, MA, USA). Reaction mixtures contained 2.5 μL diluted cDNA, 0.3 μM of each primer, and SYBR^®^ Selection Master Mix (Applied Biosystems) in a total volume of 20 μL. The sequences of each primer pair used in RT-qPCR experiments are indicated in Appendix A. Reaction mixtures were incubated for 2 min at 50 °C and then 5 min at 95 °C; this was followed by 40 amplification cycles consisting of 15 s at 95 °C and 1 min at 60 °C. Analyses of melting curves were performed immediately after the completion of the RT-qPCR. The RT-qPCR expression data were analyzed by One-way ANOVA (Nonparametric) through Dunnett post test using GraphPad Prism 5.

### 2.10. RT-PCR

The alternative splicing mechanism of *chl/mitAPX* was evaluated by RT-PCR using cDNAs from castor bean leaves and roots (see Section 2.9) and Taq platinum (Invitrogen^®^) according to the manufacturing instructions. The primer set used was S1 (5′-GTATTGTGTTAGATGGCGCT-3′), S2 (5′-GAGATGTCAACAATGCCAGAA-3′), and S3 (5′-CTGCATTTCAAATAGGTAATACT-3′).

### 2.11. miRNA Target Prediction in Castor Bean APX, MDAR, DHAR, and GR Genes

The psRNAtarget tool was used for predicting miRNA target regulation in castor bean genes, with a default score scheme [52]. Target candidates were challenged against a list of 91 castor bean miRNA sequences retrieved from the sRNAano database [53].

## 3. Results and Discussion

### 3.1. Identification and Phylogenetic Analysis of APX, MDAR, DHAR, and GR Genes

The analysis of castor bean, cassava, jatropha, and rubber tree genomes using protein sequences from arabidopsis and rice as baits allowed us to identify 59 putative sequences of genes to AsA-GSH cycle (Figure 1a). The number of APX, MDAR, DHAR, and GR genes identified in each species analyzed is indicated in Table 1 and Appendix A.

To analyze the evolutionary relationship and classify the identified sequences, phylogenetic analyses were conducted. The analysis of APX sequences shows a clear divergence among cAPX, pAPX, and chl/mitAPX genes was observed (Figure 1b). Previous work indicated that the ancestral APX was possibly a soluble enzyme targeted to the chloroplast stroma, which generates cAPX and pAPX through duplication and neofunctionalization events during land colonization by the first streptophytes. Posteriorly, in basal angiosperm, the chlAPX acquired the capacity to be dually targeted to chloroplast and mitochondria [17]. The phylogenetic analysis of the APX family reveals as the first dichotomous branching the divergence between the chl/mitAPX and cAPX/pAPX isoforms, distinguishing two main groups (Figure 1b). These data confirm the hypothesis that the cAPX and pAPX subfamilies were generated more recently through a duplication event in an ancestral non-chloroplastic/mitochondrial isoform [17,54]. 

MDAR and DHAR enzymes are also recognized to emerge as chloroplastic enzymes in photosynthetic organisms, and the major duplication and neofunctionalization events that give rise to the different subfamilies must have occurred in the streptophyte ancestor, before the divergence of charophytes [12,17]. The phylogenetic analysis of the MDAR family demonstrates the presence of three main groups, corresponding to chl/mitMDAR, pMDAR, and c/pMDAR genes (Figure 1c). As observed in the APX family, the phylogenetic analysis indicates the divergence of chloroplastic/mitochondrial isoforms in the first dichotomous branching, possibly due to the chloroplastic nature of APX and ascorbate-reducing enzymes [17]. The DHAR family is divided into two main groups, corresponding to chlDHAR and cDHAR genes (Figure 1d). The GR family is also divided into two groups, corresponding to cGR and chl/mitGR genes (Figure 1e).

It is recognized that dual-targeted proteins arose early in land plant evolution, and they are targeted simultaneously to more than one compartment through mechanisms dependent on ambiguous targeting signals or alternative transcription/translation processes. Despite the physiological role of dual targeting proteins remaining largely unknown, it is suggested that it may be related to the restriction of gene copy number or genome size [55]. The dual targeted mechanisms appear to be conserved in castor bean, cassava, jatropha, and rubber tree. While chl/mitAPX proteins are found only in angiosperms [17], the MDAR and GR dual targeted to chloroplast and mitochondria are found in different streptophyte species, such as *Physcomitrella patens*, *Picea glauca*, rice and arabidopsis [56,57], suggesting that chloroplast/mitochondria dual-targeting ability of MDAR and GR family arose earlier during land plant evolution. 

In addition to proteins dual targeted to chloroplast and mitochondria, the c/pMDAR group is located in both cytosol and peroxisomal matrix due to a weak peroxisome-targeting signal sequence in the C-terminus [12]. It is recognized that these dual-targeted proteins also emerged during the evolution of the first land plants [12,17]. A more recent duplication and neofunctionalization events in Brassicaceae generated enzymes without the peroxisome-targeting signal and specifically targeted to the cytosol [12]. Thus, in arabidopsis, AtMDAR2 (AT3G09940) and AtMDAR5 (AT5G03630), which lost the peroxisome-targeting signal, are target only to the cytosol [58,59]. 

### 3.2. Structural Organization of APX, MDAR, DHAR, and GR Genes

To investigate the relationships among the different genes encoding the APX, MDAR, DHAR, and GR isoforms in Euphorbiaceae, we compared their chromosomal locations and structural organization in castor bean and cassava, which have the genomic organization in chromosomes available. The chromosomal location and the duplication events that generate paralogous genes pairs in castor bean (Figure 2a) and cassava (Figure 2b), and the syntenic analysis of orthologous genes from these species (Figure 2c) indicates a close evolutionary relationship among the AsA-GSH cycle genes and a variable chromosomal distribution.

The analysis of paralogous gene pairs indicates that duplication events occurred mainly in APX family, which are generally more ancestral and present in all analyzed species (Table 2). While the ancient duplication events within the *APX* family occurred before the divergence of the analyzed species, the paralogous gene pairs MeAPX1 and MeAPX2, MeAPX6 and MeAPX7, and HbAPX4 and HbAPX5 appear to be generated more recently by species-specific duplication events. Only recent duplication events were verified to MDAR, DHAR, and GR paralogous gene pairs. Paralogous pair of DHAR is present only in castor bean (RcDHAR2/RcDHAR3), being the unique in tandem duplication event identified. The Ka/Ks ratios of each duplicated gene pair were generally <1, suggesting the occurrence of purifying selection. The exception is the Ka/Ks ratio to the HbAPX2/HbAPX3 pair, which is >1, indicating a positive or Darwinian selection, possibly due to the presence of an advantageous mutation. Indeed, this paralogous gene pair shows a high nonsynonymous substitution rate (Ka).

The analysis of the structural organization of castor bean, arabidopsis, and rice AsA-GSH cycle genes reveals a high degree of conservation in exon-intron structure among the angiosperm species and the sequences belonging to the same phylogenetic group (Figure 3). The cAPX, pAPX, and chl/mitAPX subfamilies show a similar exon-intron structure and the last exon of pAPX genes encodes the transmembrane domain and the peroxisome-targeting signal (Figure 3a). In RcAPX4 gene, the exons 5 and 6, which encode the heme-binding site, are absent. In angiosperms, hybrid APX enzymes lack resides or domains essential to APX activity [60]. In chl/mitAPX genes the last exon encodes the thylakoid transmembrane domain, absent in sAPX isoforms, which are targeted to chloroplast stroma and/or mitochondria matrix.

The analysis of the exon-intron structure of MDAR genes reveals a higher divergence among the subfamilies (Figure 3b). In chl/mitMDAR subfamily the first exon, absent in other subfamilies, encodes the chloroplast transit peptide. In pMDAR subfamily, the last exon is bigger and encodes the transmembrane domain and the peroxisome-targeting signal, while in c/pMDAR subfamily the last exon encodes the weak peroxisome-targeting signal, which allows these isoforms to work in both the cytosol and peroxisomal matrix [12]. Among c/pMDAR genes, this exon is not present in RcMDAR3, resulting in a shorter coding sequence without the peroxisome-targeting signal sequence.

The analysis of DHAR genes shows higher conservation of exon-intron structure of cDHAR and chlDHAR subfamilies, which differ from each other by the presence of chloroplast transit peptide coding sequence (Figure 3c). Despite this, the equivalent to exon 1 from RcDHAR1 is absent, and the exon-intro organization of AtDHAR5 is different of other cDHAR isoforms. 

In GR family, the cGR genes have a higher complex genome architecture with 16 exons. The chlGR genes contain only 10 exons, with the exon 10 encoding a chloroplast transit peptide. Among chlGR genes, OsGR3 has not equivalent to exon 1 (Figure 3d).

### 3.3. Protein Sequence Analyses of APX, MDAR, DHAR, and GR

The APX genes encode polypeptides of 211–428 amino acid residues and and molecular weight between 23.6–46.4 KDa with PI value in the interval 5.31–9.36 (Appendix A). The variations in protein properties are correlated with their respective subfamily due to the presence of transit peptides and transmembrane domains. The chl/mitAPX subfamily displays the highest instability index among APX subfamilies. Indeed, the instability of chlAPX proteins was demonstrated in different species, and it was already shown that high levels of endogenous ascorbic acid are necessary to prevent chlAPX inactivation [61].

The APX structure and catalytic mechanism have been extensively studied, and two typical structural domains of heme peroxidases were identified: the active site domain (ASD) and the heme-binding domain (HBD) [62,63]. The analysis of presented APX amino acid sequences reveals that, except for RcAPX1 and RcAPX4, these domains are highly conserved among the different APX subfamilies (Appendix A). In the chl/mitAPX subfamily, two specific organellar signature domains are found (Org-D1 and Org-D2), not present in the cAPX and pAPX subfamilies [54]. These domains are present in all chl/mitAPX, except JcAPX5, which does not show the Org-D1 (Appendix A).

This analysis of MEME motifs in APX sequences identified 15 distinct conserved motifs (Figure 3e). The cAPX, pAPX, and chl/mitAPX subfamilies show the same motif composition, with ASD in motif 2, and HBD in motif 1. The motif 11 is found only in the pAPX subfamily and corresponds to the transmembrane domain and peroxisome targeting signal. The chl/mitAPX subfamily shows an N-terminal extension, which corresponds to the chloroplast/mitochondrial transit peptide. Among chl/mitAPX sequences, only AtTAPX and OsAPX8 show a C-terminal extension with the motif 13, which corresponds to the thylakoid transmembrane domain. The sequence logos for the 15 conserved motifs are shown in Appendix A.

The analysis of MDAR sequences predicts polypeptides of 265–594 amino acid residues and molecular weight between 29.3–65.8 KDa with PI value in the interval 5.60–8.94 (Appendix A). MDAR proteins have three conserved putative amino acid domains involved in the enzymatic activity: FAD ADP-binding domain (FABD), NAD(P)H-binding domain (NBD), and FAD flavin-binding domain (FFBD) [64]. The alignment of MDAR sequences shows that FABD, NBD, and FFBD are conserved in all proteins (Appendix A). Among chl/mitMDAR subfamily, JcMDAR1 does not show the FABD and JcMDAR4 does not show both FABD and NBD. In c/pMDAR subfamily, RcMDAR3 displays only part of FABD and FFBD, JcMDAR3 shows a low conservancy in FABD, and HbMDAR6 lacks FABD.

The MEME motif analysis shows that most of the 15 distinct conserved motifs are conserved in all subfamilies (Figure 3f). The FABD is present in motif 11, NBD in motif 1, and FFBD in motif 3. The motif 14 is present only in pMDAR subfamily, and corresponds to the transmembrane domain and peroxisome targeting signal. Despite the high conservancy of protein motifs, RcMDAR3 does not show the motifs 6, 8, 5, and 7, resulting in a shorter polypeptide, and OsMDAR3 has not the motif 7. The sequence logos for all motifs identified to MDAR proteins are shown in Appendix A.

The DHAR-predicted polypeptides show a length of 183–273 amino acid residues, molecular weight of 20.6–30.3 KDa, and PI value between 5.32–8.48 (Appendix A). Among them, the chlDHAR subfamily displays the highest instability index. The DHAR proteins display two distinct domains: The Glutathione-S-Transferase N-terminal domain (GSTND) and the Glutathione-S-Transferase C-terminal domain (GSTCD) (Appendix A). These domains are conserved in the sequences analyzed, except for AtMDHAR5, which shows partial deletion in GSTND and GSTCD. 

Among the 15 distinct conserved motifs identified in DHAR family, the motif sequential order 6-3-2-5-1-4 is present in most DHAR proteins, with GSTND found in motifs 3 and 2, and GSTCD in motifs 2, 5, and 1. RcDHAR1 does not display the motifs 6 and 3, and AtDHAR5 does not display the motifs 6, 3, and 5 (Figure 3g). Consequently, the predicted RcDHAR1 and AtDHAR5 proteins are shorter, and the chloroplastic/mitochondrial transit peptide from RcDHAR1 is absent. The sequence logos for all DHAR motifs are shown in Appendix A.

The peptide length, molecular weight, and pI of GR proteins ranged between 488–560 amino acid residues, 52.7–60.4 kDa, and 5.63- 8.05, respectively (Appendix A). The average molecular weight of the chl/mitGR subfamily (group II) is slightly higher than the cGR subfamily (group I) due to the presence of an extended chloroplastic/mitochondrial transit peptide at the N-terminus of chl/mitGR proteins. Based on pI value analysis, the cGR proteins are generally acidic, while chl/mitGR proteins are basic. 

The analyses of chl/mitGR and cGR protein sequences revealed the presence of GR-specific structural domains, such as the active site domain (ASD), the NADPH binding domain (NBD), and the FAD-binding domain (FBD) (Appendix A). These domains are highly conserved among the majority of GR proteins, but not in JcGR1, in which ASD and FBD are not conserved. In cGR proteins, a cytosolic signature domain (Cyt-D) is described [65], which was present only in cGR proteins. Motif scan through the MEME tool allows the identification of 15 conserved motifs in GR proteins (Figure 3h). The majority of identified motifs is conserved in both GR subfamilies. The ASD is found in motif 6, NBD in motif 3, and FBD is divided between motifs 4 and 1. In chl/mitGR subfamily, an N-terminal extension corresponds to the chloroplast/mitochondrial transit peptide. The sequence logos for all conserved motifs identified in GR are shown in Appendix A.

### 3.4. Analysis of Alternatively Spliced mRNA Variants of Castor Bean chl/mitAPX

Previous work demonstrated that angiosperm chl/mitAPX emerged as a unique gene that generates both chloroplastic and mitochondrial isoforms by alternative splicing mechanism, which also generates the soluble and thylakoid membrane-bound enzymes [17,66]. This splicing mechanism has been demonstrated previously in different species [66]. On the other hand, in Brassicaceae, Poaceae, and Salicaceae families, individual duplication and neofunctionalization events allow individual genes encoding soluble and thylakoid membrane-bound enzymes [17,66].

The chl/mitAPX coding sequence generally contains 13 exons. The exon 12 encodes a terminal aspartic acid residue and the stop codon, and generates a soluble APX (sAPX). Alternatively, exon 13 encodes the thylakoid membrane domain and the stop codon for the thylakoidal APX isoform (tAPX). As a result of an alternative splicing mechanism, chl/mitAPX genes produce four types of mRNA variants, one tAPX and three forms of sAPX (sAPX-I, sAPX-II, and sAPX-III) (Figure 4a) [67]. Among sAPX, the sAPX-II contains a longer version of exon 11, resulting in seven additional amino acid residues and an alternative stop codon [67]. To evaluate the occurrence of alternative splicing of chl/mitAPX transcripts in Euphobiaceae, we use castor bean as a model. Here, we evaluate the read coverage of RcAPX5 sequence by RNAseq analyses of leaves and roots under control conditions (bioproject PRJNA401329). As expected, this analysis corroborates the presence of different alternative transcripts of RcAPX5 gene (Figure 4b).

The nucleotide sequences of the region of intron 11, exon 12, and exon 13 from castor bean, cassava, jatropha, and rubber tree chl/mitAPX were compared (Figure 4c). Our results indicate high conservation among these sequences and the presence of splicing regulatory elements (SRE) before exon 12 and exon 13, as previously demonstrated to different angiosperms [17]. Altogether, our data indicate that in Euphorbiaceae, the chl/mitAPX splicing is similar to that verified in other angiosperm species [66]. Among Euphorbiacea chl/mitAPX, HbAPX4 sequence shows a deletion in the intron 11′- exon 13 junction, resulting in the elimination of the SRE responsible for generating tAPX. Consequently, *HbAPX4* gene is predicted to encode only the sAPX isoforms. The phylogenetic and genomic analyses indicate that this neofunctionalization event occurred specifically in rubber tree, after a duplication event of chl/mitAPX gene of cassava and rubber tree ancestral. This divergence mechanism occurred in about 20 MYA (Table 2), resulting in two paralogous genes. While rubber tree acquired a new chl/mitAPX that encodes an exclusively soluble APX, both cassava genes are predicted to encode different APX variants by alternative splicing. 

To experimentally demonstrate the alternative splicing mechanism of chl/mitAPX, the presence of the four different splicing variants was evaluated in castor bean by RT-PCR using a primer set designed based on previous work [67]. All mRNA variants (sAPX-I, sAPX-II, sAPX-III, and tAPX-I) were detected in the leaves and roots of castor bean (Figure 4d).

### 3.5. Transcriptional Profiles of Castor Bean APX, MDAR, DHAR, and GR Genes

Among Euphorbiaceae species, castor bean has a high drought tolerance as an important trait. This plant presents a high amount of oil in the seeds [23]. As expected, the seeds production and oil quality are adversely affected by abiotic stresses, such as drought, which are highly regulated by ROS metabolism. For these reasons, the expression pattern analysis was focused on this species.

To analyze the expression pattern of RcAPX, RcMDAR, RcDHAR, and RcGR genes, we investigated their transcriptional profiles in different organs and seed developmental stages by previously published RNA-seq method (Figure 5a) [47]. Among the different subfamilies, the cytosolic isoforms are more expressed, while the peroxisomal isoforms are less expressed. In this data set, we did not find significant expressions of RcAPX1 and RcAPX4 in any tissue. The expression profile reveals that throughout seed development, the expression of ASA-GSH cycle genes is generally repressed during the transition from endosperm II-III to endosperm V-VI stage, being induced during germination.

In castor bean seeds, gluconeogenesis is the main pathway that provides the sugar energetic source to germination, being an important source of ROS mainly through the succinate production from fatty acid storage via the glyoxylate cycle [68]. In oilseeds, high amounts of hydrogen peroxide are also generated as a product of fatty acid β-oxidation [69], and the AsA-GSH cycle activity is regarded to protect oil bodies during seedling growth [70]. Thus, the activity of AsA-GSH enzymes in castor bean seeds may contribute to preventing the peroxidation of lipids storage, being related to the acquisition of drying tolerance and seed quality [71]. Previous works in different species confirmed the importance of AsA-GSH cycle enzymes during plant germination. In tobacco, the overexpression of arabidopsis pAPX (AtAPX3—AT4G35000) [72], tobacco tAPX [73], and *Salicornia brachiate* pAPX [74] improves seed germination under osmotic stresses. On the other hand, the silencing of tAPX in rice (OsAPX8) led to delayed germination [75]. The MDAR and GR genes were also demonstrated to play a central role in the antioxidant defense during seed germination. In arabidopsis, AtMDAR3 (AT3G27820) is required for seed storage oil hydrolysis and post-germinative growth [76], while AtGR2 (AT3G54660) maintains the reduced glutathione levels in maturing and germinating seeds [77].

Previous work reported a central role of the AsA-GSH cycle in the responses to different abiotic or biotic stress conditions, allowing the activation of different signaling pathways [19,20]. To understand antioxidant mechanisms related to castor bean drought response, we performed an RNAseq analysis and evaluated the APX, MDAR, DHAR, and GR expression profiles in leaves and roots of plants submitted to drought (water potential of −1.0 MPa). These data demonstrate a distinct response in leaves and roots (Figure 5b). In leaves, the drought stress reduced mainly the expression of RcAPX2, RcAPX5, RcDHAR1, and RcGR2, while RcMDAR1 and RcGR1 are the genes induced. On the other hand, the gene expression response in roots is more discreet, with repression of RcAPX4. As observed in the organ expression analysis, RcAPX1 expression was not verified in both leaves and roots, and RcAPX4 transcript was not verified in leaves. 

To access the expression profile of AsA-GSH cycle genes in castor bean submitted to different water potentials (−0.5, −1.0, and −1.5 MPa), we analyzed their expression by RT-qPCR (Figure 6). Among APX genes, RcAPX2 and RcAPX4 are repressed in leaves and roots (Figure 6b,d), while RcAPX3 is induced in both organs (Figure 6c). Despite RcAPX1 being induced in both leaves and roots, the induction in leaves occurs at −1.5 MPa, and in roots, the induction occurs early, at −0.5 MPa (Figure 6a). It is important to note that protein sequence analysis indicated that RcAPX1 and RcAPX4 could encode non-functional APXs, and, consequently, only RcAPX2 and RcAPX3 might contribute to APX activity in cytosol and peroxisomes, respectively. Thus, during drought stress, the cAPX activity appears to be repressed in both leaves and roots, while pAPX activity is induced. The expression profile of sAPX and tAPX variants of RcAPX5 is very similar, being repressed in leaves (Figure 6e,f), which can result in a decrease in chl/mitAPX activity. No changes in stromal and thylakoidal variants of RcAPX5 were verified in roots.

APX activity was previously demonstrated to be increased in leaves from castor bean exposed to drought [78] and salinity [79]. Based on our expression data, the increased APX activity in leaves during drought stress may be supported by the induction of the peroxisomal RcAPX3. Indeed, the peroxisomes are recognized as the main site of ROS production in photosynthetic tissues and have an important role during drought stress response [80,81]. In castor bean, the increased peroxisomal APX activity could control the hydrogen peroxide levels derived from photorespiration during drought stress, contributing to the high tolerance. In addition to stress tolerance, the overexpression of pAPX also improves productivity in plants subject to drought. In tobacco, the overexpression of arabidopsis pAPX led to increased fruit number and seed mass compared to non-transgenic plants [72], confirming the important role played by pAPX during osmotic stresses, when the stomata conductance and the CO_2_ supply is limited.

The MDAR genes also show a different expression pattern. The RcMDAR1 is induced in both leaves and roots at −1.5 MPa stress (Figure 6g), while RcMDAR2 is repressed at −0,5 and −1.0 MPa, but not at −1.5 MPa (Figure 6h). The RcMDAR3 expression diverges among leaves and roots, being repressed in leaves, and induced in roots (Figure 6i). The analysis of DHAR expression shows that RcDHAR1 is induced in roots at −1.5 MPa, while is repressed in leaves during all stress stages (Figure 6j). The RcDHAR2 and RcDHAR3 responses are similar in leaves and roots. The RcDHAR2 is repressed at −0.5 MPa but induced at −1.5 MPa (Figure 6k), and RcDHAR3 is repressed in all stress conditions (Figure 6l). It is important to note that RcDHAR2 displays a 5-fold induction in leaves, while it is 40-fold induced in roots. Among GR genes, RcGR1 is also repressed in all stress conditions (Figure 6m), and RcG*R2* is repressed at −0.5 and −1.0 MPa (Figure 6n). At −1.5 MPa stress, the RcGR2 is induced in roots and not changed in leaves.

To analyze the potential interaction pattern of castor bean AsA-GSH cycle genes, correlation networks were constructed using the ten proteins with the best interaction score with each APX isoform. These analyses were also used to investigate the network’s response to drought in leaves and roots, where the transcription profile of the genes identified in each association network was analyzed upon the stress treatment (Figure 7). As expected, reinforcing the connection among the enzymes from AsA-GSH cycle, the main proteins found in APX networks are the MDAR, DHAR, and GR. In addition, other proteins related to ascorbate metabolism, such as L-galactono-1,4-lactone dehydrogenase (GLDH) and ascorbate oxidase (AAO), and other proteins involved in antioxidant metabolism, such as peroxiredoxins (PRX), superoxide dismutase (SOD), glutathione peroxidase (GPX) and overexpressor of cationic peroxidases (OCP) were also verified. It is worth highlighting that the RcAPX1 and RcAPX2 networks show the same components (Figure 7a,b). The analysis of the correlation network combined with the RNAseq expression profile shows general repression in leaves during drought response, while in roots, the response is less intense. Altogether, these data indicate a distinct regulatory mechanism of the network in leaves and roots during the drought.

### 3.6. Analysis of Putative Cis-Regulatory Elements and Scanning for miRNA Targeting in APX, MDAR, DHAR, and GR Genes

To identify regulatory mechanisms of AsA-GSH cycle genes in Euphorbiaceae, cis-regulatory elements related to growth and development, and response to abiotic stress or phytohormones were searched in the promoter region of identified genes. These cis-regulatory elements were classified into three categories, including environmental conditions, phytohormones, and growth and development-responsive elements (Appendix A).

Among the environmental conditions responsive cis-regulatory elements, elements related to light, anaerobic, and drought were the main detected (Appendix A). These elements include mainly Box4, G-Box, ARE, GT1-motif, LTR, MBS, TCT-motif, I-box, and TC-rich repeats (Appendix A). Overall, these data indicate that most of the analyzed genes could be regulated during these environmental conditions. The phytohormones responsive elements include elements responsible for jasmonate (Me-JA), abscisic acid (ABA), gibberellic acid, salicylic acid (SA), and auxin (Appendix A). Among them, ABRE, CGTCA/TGCAG-motives, and TCA-element are the most abundant (Appendix A). These elements are responsive to ABA, Me-JA, and SA, which are related to different environmental stress responses. ABA exerts a central role in the regulation of gene expression in response to osmotic stress, such as drought and high salinity, and induces ROS production [82] and the activities of SOD, CAT, and APX enzymes [83,84,85,86]. ABA treatment increases the APX expression in different plants, such as rice [87], pea [88], and sweet potato [89]. Me-JA and SA are also related to induce ROS production and the activities of antioxidant enzymes [90,91,92] and regulate the stomata closure in response to drought [93,94]. Thus, our data suggest that the expression of ASA-GSH cycle genes may be regulated by these phytohormones, contributing to drought tolerance.

The growth and development responsive elements consist of elements related to flavonoid metabolism, meristem expression, endosperm expression, circadian control, and cell cycle regulation (Appendix A), such as CAT-box, GCN4_motif, MBSI, AACA_motif, circadian and MSA-like (Appendix A). Interestingly, the presence of these elements varies among the different species analyzed, indicating that the AsA-GSH cycle genes may be involved in species-specific growth and development processes.

The search of small regulatory RNAs could target the castor bean AsA-GSH cycle genes, allow us to identify 15 miRNAs targeting four genes: RcAPX5, RcMDAR2, RcDHAR2, and RcGR1 (Appendix A). The identified miRNAs belong to six different families: miR156, miR6445, miR167, miR171, miR390, and miR159.

These miRNAs have been reported to target different plant development and stress-responsive genes. The miR156 is predicted to target RcAPX5 (RcAPX5s and RcAPX5t variants) (Appendix A) and is recognized as the master regulator of the vegetative phase and the transition to the adult phase [95,96]. Here, we identified three miRNAs predicted to target RcMDAR2: miR6445, miR167, and miR171 (Appendix A). miR6445 was demonstrated to regulate the expression of NAC transcription factors in *Populus trichocarpa*, controlling a variety of plant developmental processes [97]. The miR167 regulates the expression of Auxin Response factor (ARF) family members AtARF6 and AtARF8, essential for fertility of both ovules and anthers in arabidopsis [98], and miR171 regulates shoot meristem activity and phase transition being able to repress the HAIRYMERISTEM (HAM) family genes [99]. It is important to note that miR6445 shows a lower expectation value, indicating that it is most likely to regulate the expression of analyzed genes (Appendix A). The miRNA miR390, in addition to being predicted to target RcDHAR2 (Appendix A), binds to arabidopsis ARGONAUTE 7 (AtAGO7), trigging siRNA biogenesis and led to regulation of AtARF3 and AtARF4 expression [100]. The miR159, which is predicted to target RcGR1 (Appendix A), was already demonstrated to target MYB33 and MYB101 transcription factors [101]. These transcription factors have been reported to exert an important role in ABA signaling and drought stress response [102]. It is important to note that the identification and characterization of miRNAs in the Euphorbiaceae family are very limited, and the expression pattern and the capacity of these predicted miRNAs to target and regulate the expression of AsA-GSH genes was not experimentally demonstrated. Thus, further functional analyses are required to characterize the biological role of identified miRNAs in the regulation of the AsA-GSH cycle genes during development and environmental stress. Experiments such as miRNAseq and a comparison of miRNA-vs-mRNA expression may provide new evidence of this regulation.

## 4. Conclusions

In this work, the APX, MDAR, DHAR, and GR gene families from castor bean, cassava, jatropha, and rubber tree were identified, annotated, and their evolutionary history was analyzed. The expression profile of the AsA-GSH cycle genes from castor bean was evaluated in plants exposed to drought stress, revealing possible mechanisms that may explain castor bean’s high tolerance to water stress. The identification of genes related to ROS metabolism in castor bean and the evaluation of their expression in response to drought may contribute to our general understanding of antioxidant mechanisms, which is essential for physiological adaptation to dehydration. The presented results provide a foundation for the validation of the precise role of AsA-GSH cycle genes, allowing the identification of new biotechnological targets for the genetic improvement not only of species of the Euphorbiaceae family, but also of other plant species of economic interest.

## Figures and Tables

**Figure 1 biology-12-00019-f001:**
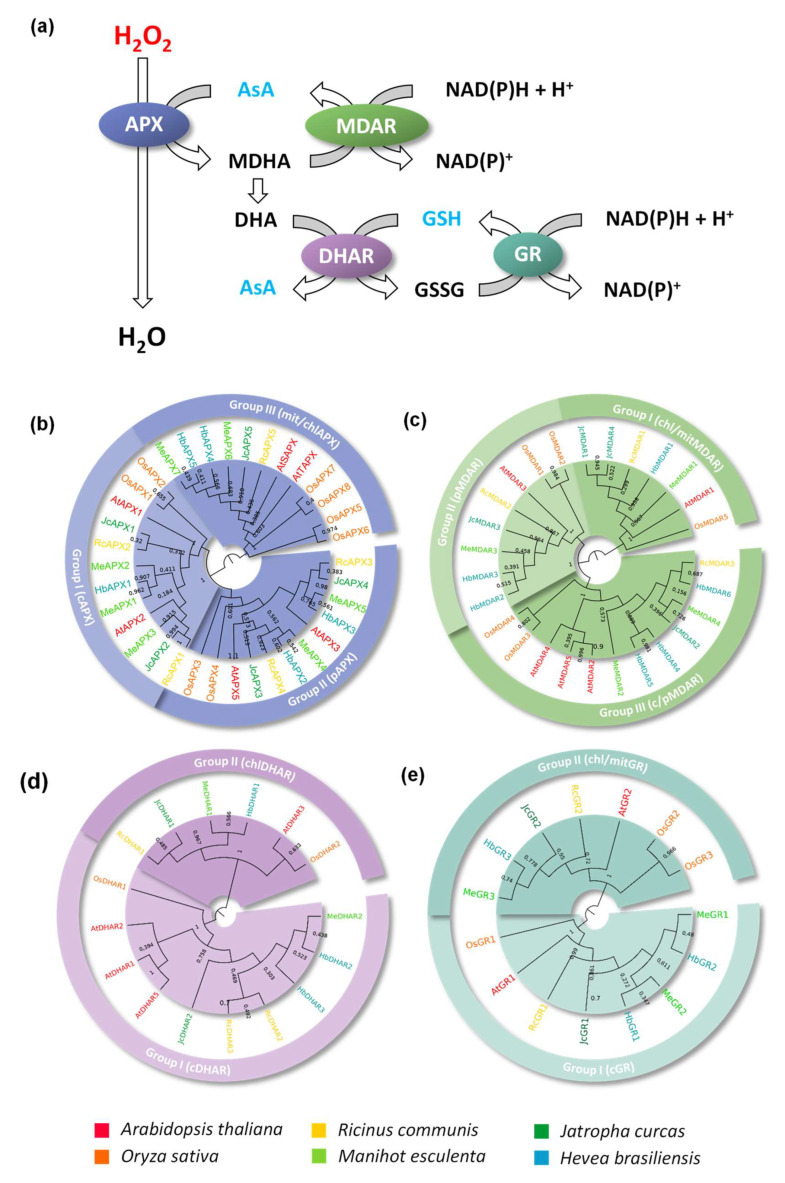
APX, MDAR, DHAR, and GR families in *Ricinus communis* (Rc), *Manihot esculenta* (Me), *Jatropha curcas* (Jc), *Hevea brasiliensis* (Hb), *Arabidopsis thaliana* (At) and *Oryza sativa* (Os). (**a**) Schematic representation of ascorbate-glutathione cycle. Phylogenetic analysis of APX (**b**), MDAR (**c**), DHAR (**d**), and GR (**e**) proteins. The phylogenetic relationships among analyzed proteins were reconstructed using the neighbor-joining method under the best model selection in MEGA 11 software with 1000 replicates of bootstrap statistics. A total of 36, 27, 16, and 15 protein sequences, respectively, were included in the analyses, and ambiguous positions were removed from the alignments. The posterior probabilities are discriminated above each branch.

**Figure 2 biology-12-00019-f002:**
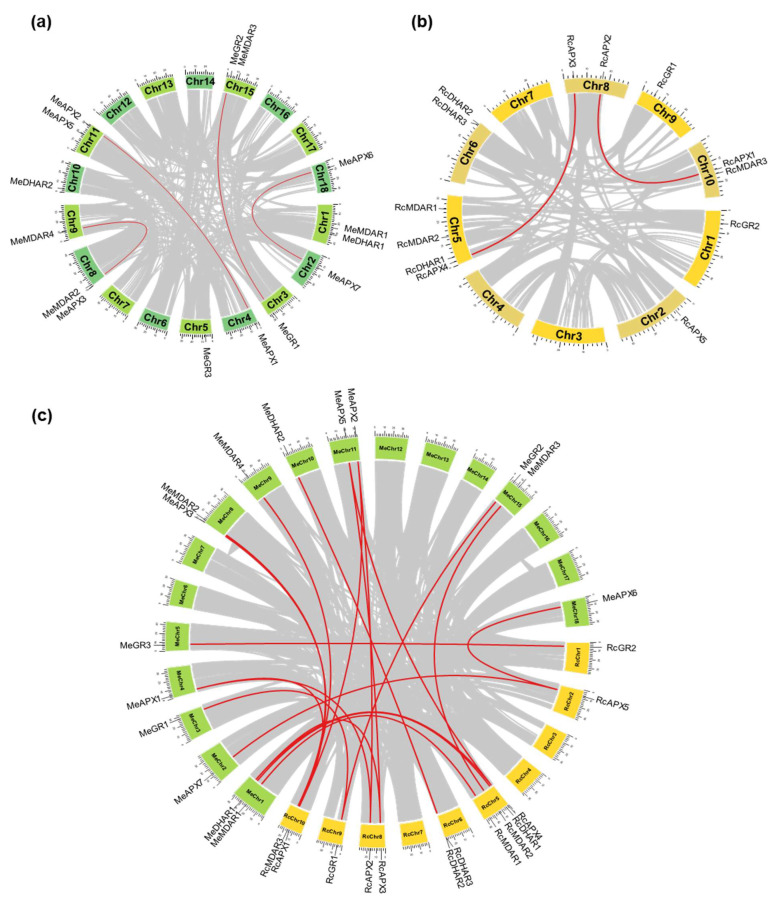
Chromosomal positions and inter-chromosomal groups of duplicated APX, MDAR, DHAR, and GR genes pairs in *Ricinus communis* (**a**) and *Manihot esculenta* (**b**). Gray lines in the background demonstrate all syntenic blocks and the red lines exhibit the segmental or tandem duplication network zones. The approximate location of APX, MDAR, DHAR, and GR genes is marked with black lines outside the chromosome names. The syntenic analysis among the *Ricinus communis* and *Manihot esculenta* genomes is indicated in (**c**).

**Figure 3 biology-12-00019-f003:**
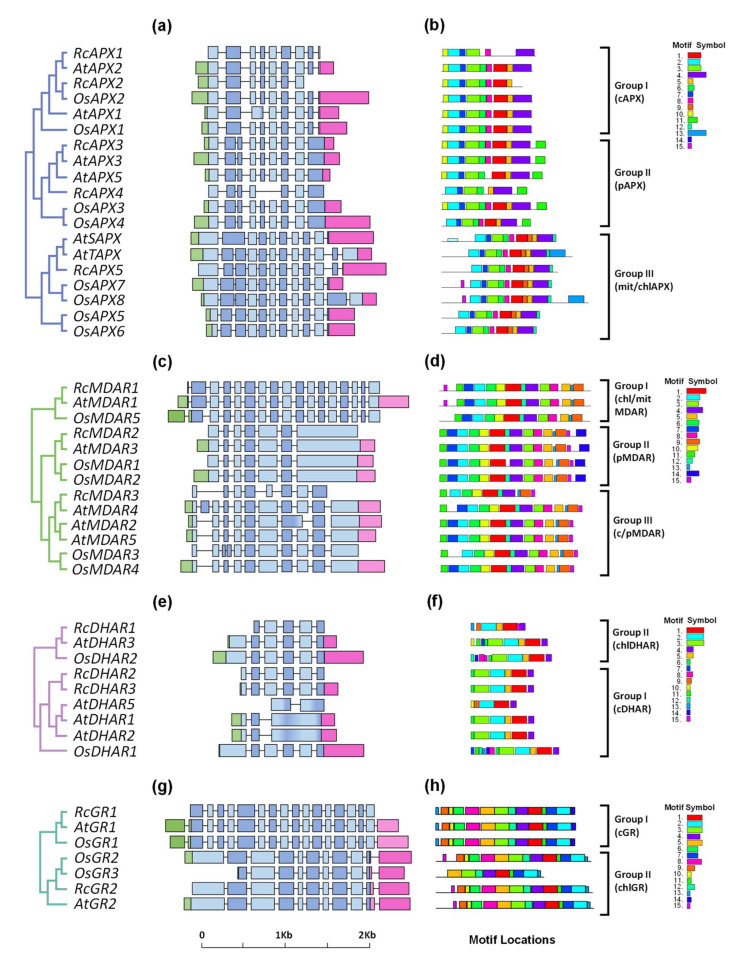
Exon-intron structure and conserved motifs analyses of APX, MDAR, DHAR, and GR families in *Ricinus communis* (Rc), *Arabidopsis thaliana* (At), and *Oryza sativa* (Os). (**a**,**e**) APX; (**b**,**f**) MDAR; (**c**,**g**) DHAR; and (**d**,**h**) GR. The phylogenetic relationships among analyzed proteins were reconstructed using the neighbor-joining method under the best model selection in MEGA 11 software with 1000 replicates of bootstrap statistics (panels **a**–**d**). For all genes represented, black lines represent introns and the lengths of exons are exhibited proportionally. All 15 conserved motifs in APX, MDAR, DHAR, and GR proteins were identified by MEME software and indicated by a colored box (panels **e**–**h**). The lines represent the non-conserved sequences, and the length of motifs in each protein is presented proportionally.

**Figure 4 biology-12-00019-f004:**
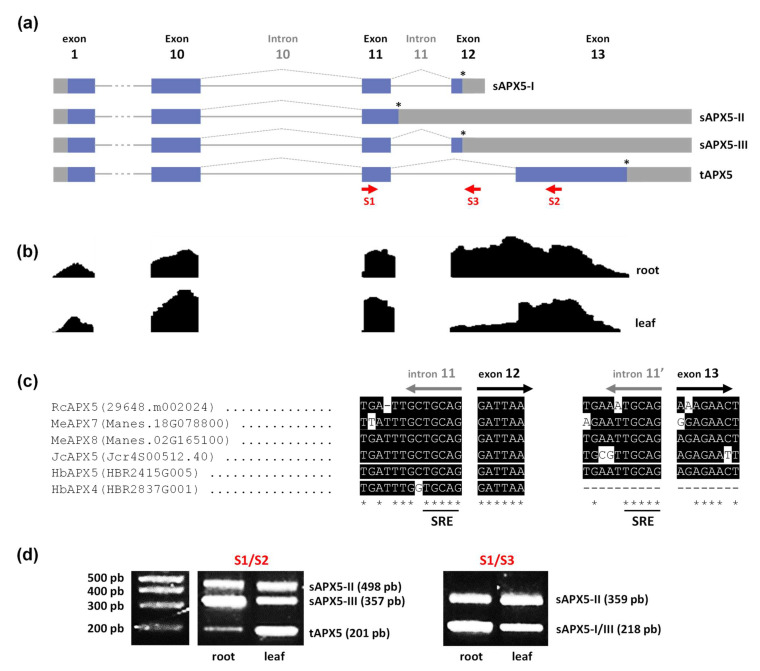
Alternative splicing from Euphorbiaceae *chl/mitAPX*. (**a**) Diagram of alternative splicing pattern producing soluble (sAPX) and thylakoidal APX (tAPX) mRNAs. Exon regions are shown as boxes and introns as lines. The gray boxes indicate 5′ and 3′ untranslated regions. The (*) indicates the stop codon from each mRNA and the red arrows indicate the primers used to confirm them. (**b**) RNAseq read-density values of RcAPX5 exons in leaves and roots under control conditions. (**c**) Comparison of the nucleotide sequences of the 3′-terminal region of intron 11 and 5′-terminal region of exon 12 and exon 13 of chl/miAPX genes in *Ricinus communis* (Rc), *Manihot esculenta* (Me), *Jatropha curcas* (Jc), *Hevea brasiliensis* (Hb), indicating the presence of splicing regulatory elements (SRE). (**d**) Experimental confirmation of castor bean sAPX and tAPX mRNAs by RT-PCR in roots and leaves.

**Figure 5 biology-12-00019-f005:**
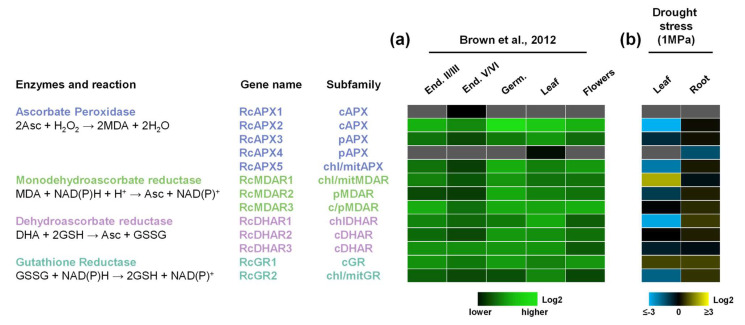
Expression profile of APX, MDAR, DHAR, and GR genes in castor bean. (**a**) Heatmap showing the expression pattern of AsA-GSH cycle genes in seeds during different developmental stages, such as endosperm II-III (End. II-III), endosperm V-VI (End. V-VI), and Germinating (Germ.), in leaves and male flowers from castor bean. The color scale represents log2 expression values, expressed in a green color scale. (**b**) Heatmap showing the expression pattern of AsA-GSH cycle genes in leaf and root in response to drought stress. The color scale represents log2 expression values, expressed in a blue to yellow color scale, relative to the expression values under control conditions.

**Figure 6 biology-12-00019-f006:**
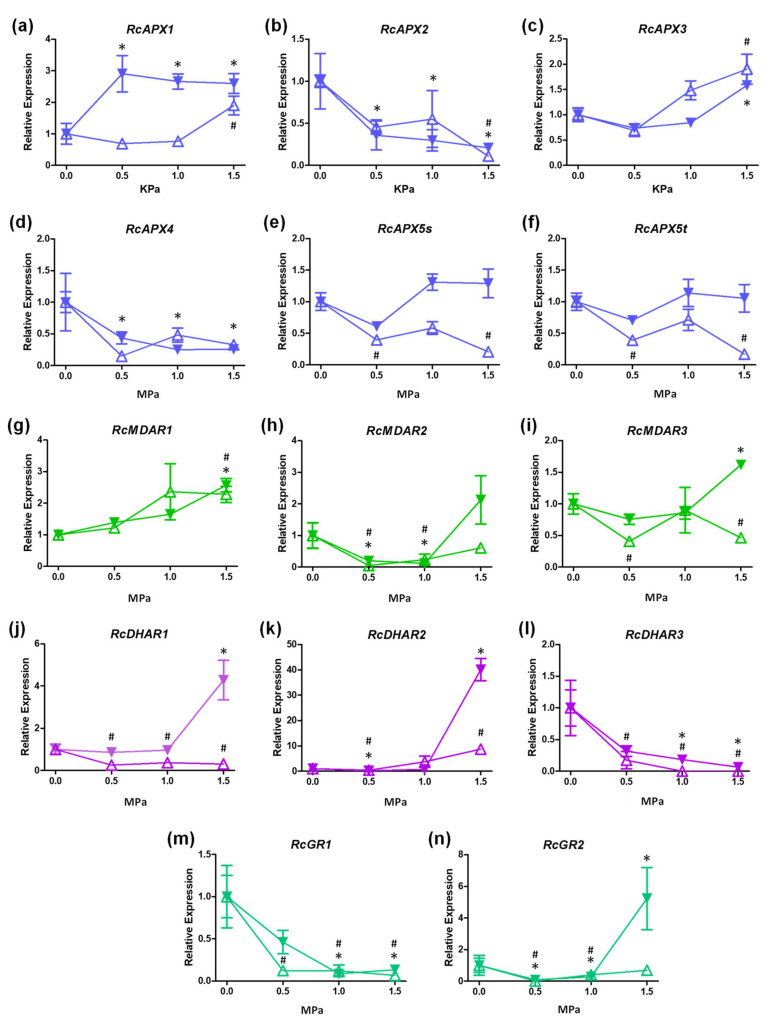
Expression profile APX, MDAR, DHAR, and GR genes in leaves and roots of castor bean submitted to drought stress. Relative expression of RcAPX1 (**a**), RcAPX2 (**b**), RcAPX3 (**c**), RcAPX4 (**d**), RcAPX5s (**e**), RcAPX5t (**f**), RcMDAR1 (**g**), RcMDAR2 (**h**), RcMDAR3 (**i**), RcDHAR1 (**j**), RcDHAR2 (**k**), RcDHAR3 (**l**), RcGR1 (**m**), and RcGR2 (**n**). Open up-triangles represent leaves and closed down-triangles represent roots. Data have been presented as a mean expression, considering the control condition as a reference. Bars represent standard error, (#) and (*) represent statistical difference between samples and control in leaves and roots, respectively, according to Tukey’s test (*p* < 0.05).

**Figure 7 biology-12-00019-f007:**
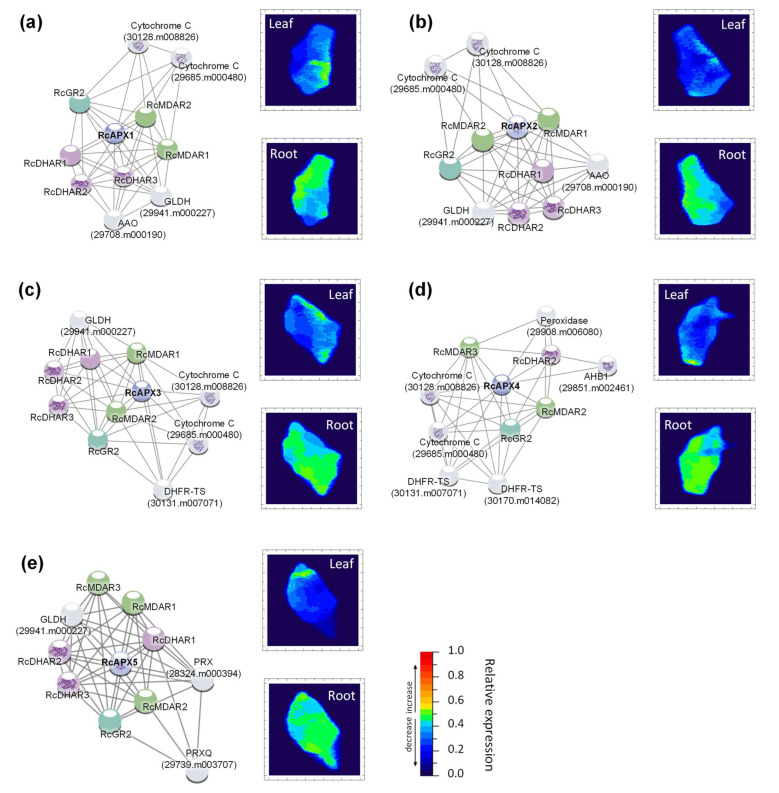
Predicted interaction partners of castor bean APX proteins. (**a**) RcAPX1, (**b**) RcAPX2, (**c**) RcAPX3, (**d**) RcAPX4, (**e**) RcAPX5. The heat maps demonstrate gene expression in response to drought in leaf (up) and root (down). The color scale at each heat map varies from dark blue to red representing the scale of the relative expression levels to control. AAO (Ascorbate oxidase); AHB1(Non-symbiotic hemoglobin); DHAR (Dehydroascorbate reductase); DHFR-TS (Bifunctional dihydrofolate reductase-thymidylate synthase); FBP (Fructose bisphosphatase); GLDH (L-galactono-1,4-lactone Dehydrogenase); GR (Glutathione reductase); MBF1B (Multiprotein-bridging factor 1b); MDAR (Monodehydroascorbate reductase); OCP (Overexpressor of cationic peroxidase); PPIA (peptidylprolyl isomerase); PRX (Peroxiredoxin); PRXQ (Peroxiredoxin Q).

**Table 1 biology-12-00019-t001:** Number of APX, MDAR, DHAR, and GR genes in Arabidopsis thaliana, Oryza sativa, Ricinus communis, Manihot esculenta, Jatropha curcas, and Hevea brasiliensis.

	*Arabidopsis thaliana*	*Oryza sativa*	*Ricinus communis*	*Manihot esculenta*	*Jatropha curcas*	*Hevea brasiliensis*
*APX*	6	8	5	7	5	5
*MDAR*	5	5	3	4	4	6
*DHAR*	4	2	3	2	2	3
*GR*	2	3	2	3	2	3

**Table 2 biology-12-00019-t002:** Ka/Ks analysis and divergence time between the duplicated APX, MDAR, DHAR, and GR gene pairs in *Ricinus communis* (Rc), *Manihot esculenta* (Me), *Jatropha curcas* (Jc) and *Hevea brasiliensis* (Hb). Ka. Non-synonymous substitution rate; Ks. Synonymous substitution rate; MYA. Million years ago.

Family	Group	Gene 1	Gene 2	Type	Ka	Ks	Ka/Ks	Date (MYA)
APX	I	RcAPX1	RcAPX2	Segmental	0.3121	3.8168	0.0818	235.6
		MeAPX1	MeAPX2	Segmental	0.0772	0.3850	0.2006	23.8
		JcAPX1	JcAPX2	Segmental	0.1645	2.5914	0.0635	160.0
	II	RcAPX3	RcAPX4	Segmental	0.2033	1.1992	0.1695	74.0
		MeAPX4	MeAPX5	Segmental	0.1685	1.5858	0.1063	97.9
		JcAPX3	JcAPX4	Segmental	0.1436	1.7389	0.0826	107.3
		HbAPX2	HbAPX3	Segmental	2.6165	1.9921	1.3135	123.0
	III	MeAPX6	MeAPX7	Segmental	0.0765	0.3804	0.2011	23.5
		HbAPX4	HbAPX5	Segmental	0.0542	0.2556	0.2119	15.8
MDAR	I	JcMDAR1	JcMDAR4	Segmental	0.7604	1.3866	0.5484	85.6
	II	HbMDAR2	HbMDAR3	Segmental	0.0388	0.2061	0.1882	12.7
	III	HbMDAR4	HbMDAR5	Segmental	0.0061	0.0098	0.6199	0.6
DHAR	I	RcDHAR2	RcDHAR3	Tandem	0.0927	0.3451	0.2685	21.3
GR	I	MeGR1	MeGR2	Segmental	0.0934	0.3710	0.2516	22.9
		HbGR1	HbGR2	Segmental	0.0330	0.2528	0.1304	15.6

## Data Availability

The data presented in this study are available in Appendix A.

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
