# Peer review of "Ascorbate-Glutathione Cycle Genes Families in Euphorbiaceae: Characterization and Evolutionary Analysis"

_biology, 2022, doi:10.3390/biology12010019_

Round 1

Reviewer 1 Report

Castor Bean is an important crop worldwide. This manuscript identified Ascorbate-Glutathione Cycle Gene Families. The authors also found some genes in this family could respond the drought stress. The manuscript presents an interesting insight the function of Ascorbate-Glutathione Cycle Gene Families for drought stress in castor bean.

Nevertheless, I have some minor comments for the manuscript.

1.      The author should provide the software, software version and parameters for each analysis in the method.

2.      The authors should provide more evidence for the miRNA-vs-mRNA expression, such as miRNA-seq and RNA-seq data.

Author Response

The manuscript has been modified according to the reviewer's suggestions (tracked in red) and brief explanations of the changes made in response to each comment are listed in the comments section.

1 - The author should provide the software, software version and parameters for each analysis in the method.

A: We thoroughly analyzed the methods section to be sure that all the version and parameters were present, including the new ones requested by Reviewer #2 (lines 235-237; 261-263). The missing information regarding the ViaComplex software (V.1.0) was included in the new version of the manuscript (line 226).

2 - The authors should provide more evidence for the miRNA-vs-mRNA expression, such as miRNA-seq and RNA-seq data.

A: We agree with the reviewer that is  very important to compare miRNA-vs-mRNA expression and that the capacity of the predicted miRNAs to directly regulate the expression of AsA-GSH genes in response to drought was not experimentally tested. However, at the present moment, our goal was to identify miRNAs that could be involved in the regulation of the AsA-GSH cycle genes. Further functional analyses are required to characterize the biological role of identified miRNAs in this regulation, especially in response to drought. The strategy adopted and its limitations have been included in the new version of the manuscript (817-824).

Reviewer 2 Report

General comments

The manuscript is too long and it is very hard to follow. I would say that it should be reduced in length by at least 40%. Also various changes in the content should be thoroughly revised and adjusted.

Specific comments

Title: The title of the manuscript is not consistent with the objective of the study.

Abstract

The main findings and impacts of the study need to be better explained.

Introduction

I do not consider the inclusion of Figure 1 in the introduction to be pertinent. It is also necessary to better focus the state of the art on drought stress. The scope and impact of the study is not clearly defined.

Materials and methods

The subsections must be numbered

Explain better the experimental design (completely randomized? factorial? the variables measured? Why only castor bean plants? Explain better the methodology of drought stress measurement.

Statistical analysis of drought stress parameters were not included

Results

Reduce the length of paragraphs, there is a lot of redundancy of ideas and written lines.

Reduce the length of paragraphs, there is a lot of redundancy of ideas and written lines.

There are many quotes included in the results section which appear to be discussing, but in reality they are presented later, in the discussion section. Correct this.

Prioritize the most relevant figures, and secondary information figures should be sent to supplementary material.

The goal of this study was to identify and initiate the functional characterization of APX, MDAR, DHAR and GR genes in Ricinus communis, Jatropha curcas, Manihot esculenta and Hevea brasiliensis. But, the results focus mainly on castor bean. Explain better

Discussion

Lines 909:835: Repeat results and no citations are included to dispute the assertions made.

The discussion on drought stress should be strengthened. The conclusions and implications of the study are also lacking.

Better organize the thread of the discussion according to how the results were presented. It is difficult to follow the reading

References

The references are too many for a research article. Prioritize the most relevant ones. 

Author Response

The manuscript has been modified according to the reviewer's suggestions (tracked in red) and brief explanations of the changes made in response to each comment are listed in the comments section.

General comments

1 - The manuscript is too long and it is very hard to follow. I would say that it should be reduced in length by at least 40%. Also various changes in the content should be thoroughly revised and adjusted.

A: As suggested by the reviewer, we performed a deep revision and re-organization the manuscript.  "Results" and "Discussion" sections have been merged to try to make the text as dynamic as possible, making it easier to follow. Despite the addition of new discussions required by both reviewers, we were able to reduce the text size by more than 2000 words.

Specific comments

2 - Title: The title of the manuscript is not consistent with the objective of the study.

A: We agree with the reviewer and the title was changed to: Ascorbate-Glutathione Cycle Genes Families in Euphorbiaceae: Characterization and Evolutionary Analysis (lines 1-2)

Abstract

3 - The main findings and impacts of the study need to be better explained.

A: The abstract has been edited in order to follow the reviewer's suggestions, respecting the limitation in character numbers from the journal (lines 37-53).

Introduction

4 - I do not consider the inclusion of Figure 1 in the introduction to be pertinent.

A: Figure 1 has been transferred to the Results and Discussion section.

5 - It is also necessary to better focus the state of the art on drought stress.

A: In the new version of the manuscript, an additional paragraph describing the relevance of drought stress was added (Lines 136-142). Additionally, a discussion about plant response to drought stress regarding ROS metabolism was included in the “Results and Discussion” section (686-710).

6 - The scope and impact of the study is not clearly defined.

A: To describe clearly the impact of the study in the Introduction, its last paragraph has been modified (lines 149-165).

Materials and methods

7 - The subsections must be numbered

A: As suggested, the material and methods subsections were numbered (167-274).

8 - Explain better the experimental design (completely randomized? factorial? the variables measured? Why only castor bean plants? Explain better the methodology of drought stress measurement.

The experimental design and drought stress measurements such as the way of sowing, the pot volumes, growth time, photoperiod, onset of drought stress, water potential measurements, number of plants used and how plant materials were collected, have been included in “Material and Methods” (subsection 2.8). The rationale of using only castor beans in the drought stress analysis has been included in the new version of the manuscript (lines 606-611). Our laboratory has focused into the analysis of castor bean response to drought due its high tolerance and since this species has a special importance in Brazilian agriculture and biotechnological application of vegetal oils.

9 - Statistical analysis of drought stress parameters were not included

A: This information was included in the new version of the manuscript (lines 235-237). We also included the parameters of the statistical analyses of the RT-qPCR experiments (lines 261-263).

Results

10 - Reduce the length of paragraphs, there is a lot of redundancy of ideas and written lines.

A: The manuscript has been revised to be reduced and avoid the redundancy of ideas, resulting in a reduction of more than 2000 words.

11 - There are many quotes included in the results section which appear to be discussing, but in reality they are presented later, in the discussion section. Correct this.

A: We agree with the reviewer's criticism and merged the “Results” and “Discussion” sections to solve this point. In the new version of the manuscript, the results are discussed right after being presented.

12 - Prioritize the most relevant figures, and secondary information figures should be sent to supplementary material.

A: The former figures 8 and 9 have been moved to the supplemental material (new Supplementary Figures  S9 and S10).

13 - The goal of this study was to identify and initiate the functional characterization of APX, MDAR, DHAR and GR genes in Ricinus communis, Jatropha curcas, Manihot esculenta and Hevea brasiliensis. But, the results focus mainly on castor bean. Explain better

A: Although part of our work was to identify and initiate the functional characterization of APX, MDAR, DHAR and GR genes in Ricinus communis, Jatropha curcas, Manihot esculenta, and Hevea brasiliensis, expression analyses have been focused on castor bean. As stated before, to clarify the rationale of the use of castor bean in the drought stress analysis was included in the new version of the manuscript (lines 606-611).

Discussion

14 - Lines 909:835: Repeat results and no citations are included to dispute the assertions made.

The discussion on drought stress should be strengthened.

A: We tried to abolish this situation by merging Results and Discussion in a single section and all the citations were reviewed. As suggested, the discussion of ascorbate glutathione cycle genes expression during drought stress was strengthened in the new version of the manuscript (Lines 685-710).

15 - The conclusions and implications of the study are also lacking.

A: A conclusion section was added in the new version of the manuscript (lines 826-838).

16 - Better organize the thread of the discussion according to how the results were presented. It is difficult to follow the reading

A: We merged the "results" and "discussion" sections to try to make the text as dynamic as possible, making it easier to follow. In the new version of the manuscript, the results are discussed right after being presented.

References

17 - The references are too many for a research article. Prioritize the most relevant ones.

A: Despite new discussions being added, we were able to reduce the text size and the number of citations from 150 to 110.

Round 2

Reviewer 2 Report

Most of the comments were well-directed, however, the paper is still too long and makes it very difficult to read. I think it should be shortened further.

Author Response

To follow the reviewer's suggestions to reduce the manuscript size and improve the paper, we edited the manuscript to once again reduce once its size (more than thousand words). The manuscript was not massively shortened because some additions requested previously by reviewers to explain your points have been made.